# Vulnerability of Maize Farming Systems to Climate Change: Farmers' Opinions Differ about the Relevance of Adaptation Strategies

**Marine Albert [1,\*], Jacques-Eric Bergez [1], Magali Willaume [2] and Stéphane Couture [3]**

[1] INRAE, UMR1248 AGIR, Université de Toulouse, F-31320 Castanet-Tolosan, France; jacques-eric.bergez@inrae.fr

[2] INRAE, UR MIAT, Université de Toulouse, F-31320 Castanet-Tolosan, France; magali.willaume@toulouse-inp.fr

[3] INPT ENSAT, UMR1248 AGIR, Université de Toulouse, F-31320 Castanet-Tolosan, France; stephane.couture@inrae.fr

**\*** Correspondence: marine.albert@inrae.fr

**Abstract:** Climate change has negative impacts on maize cultivation in southwestern France, such as soil erosion and water stress. The vulnerability of maize farming systems to climate change must be assessed before considering potential adaptation strategies. This study focused on eliciting and understanding criteria that maize growers use to assess the vulnerability of their farming systems to climate change. To this end, we surveyed maize growers in two consecutive stages: a qualitative stage, to elicit vulnerability criteria, and a quantitative stage, to test the genericity of criteria related to the adaptation strategies. The qualitative stage identified 144 criteria that farmers used to assess vulnerability to climate change, while the quantitative stage showed that farmers' opinions about the adaptation strategies differed. Many factors explained these differences, including structural (e.g., soil type) and psychological factors (e.g., interest in agroecology). Our typology of farmers revealed that their interest in agroecology and technology, as well as their perceptions of the risks of climate change and their attachment to their production systems, influence the type of adaptations they identify as relevant (i.e., intensification strategies, slight adjustments or agroecological innovations). Farmers' perceptions should be considered when providing individual advice and assessing vulnerability, by including criteria related to their psychological characteristics.

**Keywords:** evaluation; vulnerability; maize grower; climate change; farmers' perceptions; adaptation

## 1. Introduction

Farmers must increasingly address the global increase in temperature and annual variations in rainfall [1]. In France, a 6 °C increase in the mean summer temperature is expected by the end of the 21st century; although annual variations in rainfall are not expected, large monthly variations are predicted [2,3]. Higher temperatures will increase soil evaporation and crop transpiration, which will increase water demand and consumption, which will in turn influence the surface water balance and the potential for severe drought [4–6]. In addition to thermal and hydric stresses, the increased frequency and intensity of extreme climatic events can result in soil erosion due to floods, droughts or storms [6]. Droughts would have a particularly severe impact on summer-irrigated crops. In southwestern France, maize cropping systems already suffer from these climate phenomena [7,8]. Consequently, farmers must adapt their farming systems to reduce their farms' vulnerability to climate change [9]. Farmers have access to many adaptation strategies [10], such as changing varieties or crop species, changing sowing density and modifying the schedule of cropping operations [7,11–15]. However, adaptation decisions can fail, either by not achieving the objectives, or worse, by increasing vulnerability [16].

Therefore, the vulnerability of maize farming systems must be assessed before considering adaptation strategies to address climate change. More specifically, a better understanding of maize farming systems and the determinants of adaptive capacity can allow public policies and agricultural advising to enhance the agroecological transition.

The Intergovernmental Panel on Climate Change (IPCC) defines climate change vulnerability as "the degree to which a system is susceptible to, and unable to cope with, adverse effects of climate change, including climate variability and extremes" [1]. Vulnerability is a function of (i) exposure (i.e., characterized by the intensity, frequency and duration of perturbations), (ii) sensitivity (i.e., degree to which the exposure influences the system) and (iii) adaptive capacity (i.e., ability to implement adaptations to address perturbations) [17]. This conceptual framework combines biophysical and socio-economic factors to determine vulnerability [18].

The need to assess vulnerability has already been discussed [19–21] but is rarely rendered operational due to its complexity. Two main approaches are used to assess the vulnerability, or the related concept of resilience, of agricultural systems [22]: (i) the quantitative analysis of agricultural system outputs, in which vulnerability is assessed by studying dynamics related to the perturbations of agricultural outputs, such as yield or economic net return [20,23,24], and (ii) the quantitative or qualitative evaluation of predefined properties associated with vulnerability or resilience [25–27]. In the latter approach, vulnerability is assessed by considering the properties of the system identified using expert knowledge or the literature [28,29].

Most vulnerability and resilience assessment studies are based on a dynamic performance approach. However, this approach is limited to easily recordable components, and often focuses on only one type of performance, mainly yield [22,30,31], whereas assessing multiple types of performance is essential [19]. Moreover, these studies rarely focus on assessing vulnerability at the farm scale, even though all the dimensions of a farming system must be considered. Identifying the determinants of vulnerability to climate change requires a systemic approach, since vulnerability is a complex problem and can be influenced by diverse factors such as crop management and the financial status of the farm. Moreover, decisions about adaptation strategies are made at the farm level [30,32]. The predefined property approach can focus on a broader scale by using indicators to consider different components of agricultural systems, including adaptive capacity [28]. Despite this advantage, few agricultural studies have used a predefined property approach [33].

Many studies considered only the dimensions of exposure and sensitivity when assessing climate change vulnerability, ignoring the adaptive capacity of farmers [34]. However, Marshall et al. [35] showed that farmers' perceptions of their skills, their satisfaction with the adaptations they implement and their willingness to change, strongly influence their adaptive capacity. Consequently, farmers' perceptions will influence the vulnerability of their farms [35]. Previous studies have stressed the importance of cognition (i.e., the administration and implementation of information) on vulnerability [36–38]. For example, Marshall et al. [35] showed that less vulnerable farmers were well integrated into social networks. In the literature, there are several empirical studies studying farmers' adaptive capacity specifically regarding climate change impacts [39–43]. Some of these studies use individual interviews with farmers to obtain their perceptions about climate change [39,40,43], while other studies base their data collection on participative methods such as focus groups [41,42]. The data analysis conducted to identify the determinants of adoption can be based on Pearson correlation [40], mathematical models [39] such as the logit model [43], and factorial analysis using PCA and ANOVA [41]. The results of these studies show that adaptive capacity is influenced by the individual characteristics of the farmers, such as access to extension services [39,42], perceptions of risks [40,43], access to information [39,43], or the level of knowledge [39,41].

Ultimately, studying adaptive capacity is essential to identify criteria for assessing the vulnerability of farming systems.

Most assessment methods are based on an objective approach (i.e., external judgements [44]) to identify performance indicators that measure the dynamics of vulnerability or identify properties that influence vulnerability. Although scientists have taken responsibility for designing these indicators, farmers are able to identify the determinants of vulnerability on their own farms [44] and can understand their situations [45] in relation to climate change. Using the determinants of vulnerability identified by farmers will help legitimize this set of indicators [44,46]. Moreover, considering farmers' personal characteristics (i.e., cognitive and psychological factors) when assessing farm vulnerability is crucial for advisors to provide specific guidelines based on each farmer's situation [35]. Jones [44] highlighted that subjective approaches can consider farmers' knowledge and experience of resilience, along with the factors that contribute to them, and can complement objective approaches.

This study aimed to identify and understand the criteria that maize growers use to assess the vulnerability of their farming systems to climate change. The criteria for assessing vulnerability were elicited (i) at the farm level (ii) to include the adaptive-capacity dimension of vulnerability, using (iii) a predefined property approach and (iv) a subjective approach based on farmers' perceptions. We conducted a two-stage survey with maize growers: one group of farmers was surveyed to elicit vulnerability criteria, and another group was surveyed to test the genericity of adaptation strategy criteria and understand farmers' opinions about the relevance of adaptations. We assumed that farmers' psychological and cognitive factors would explain their opinions. This study will help understand farming with a more systemic approach, by knowing to what extend the farmer's subjectivity needs to be considered when investigating vulnerability to climate change.

## 2. Materials and Methods

### 2.1. Conceptual Framework

We developed a conceptual framework that combines the Drivers–Pressures–State-Impacts–Responses (DPSIR) model developed by Smeets et al. [47] and the three dimensions of vulnerability defined by the IPCC (i.e., exposure, sensitivity and adaptive capacity) [17]. The DPSIR model represents causal interactions between a system and its environment: a driving force (D) creates pressure (P) on a system in a given state (S), which creates impacts (I). In response to these impacts, the system responds (R) with adaptations [47,48].

We developed the following framework to explore farm vulnerability to climate change (Figure 1): climate change variables are driving forces (D) (e.g., rainfall) that produce pressures (P) (e.g., excess water) that influence the state of cropping systems (S). This degradation of cropping systems has impacts on the farm (I) (e.g., soil erosion) that require adaptation strategies (R) (e.g., reduction in soil tillage). We added a supplementary component to the DPSIR model in our framework, since implementing adaptation strategies requires the resources of the farm (e.g., availability of equipment, financial situation) and of the farmer (e.g., knowledge, perceptions) (F). Along with climate change pressures, other driving forces can pressure the system, such as market volatility, regulations and citizens' opinions. We considered only the negative aspects of climate change in our conceptual framework, although it could include positive aspects (e.g., increasing $CO_2$ concentrations increase plant growth).

### 2.2. Overview of the Survey Design

We used an empirical approach that had two consecutive stages to include qualitative and quantitative approaches (Table 1). The qualitative stage aimed to elicit vulnerability criteria from a group of expert farmers. We considered "criteria" instead of "indicators" since we focused on identifying the determinants of vulnerability and not on measuring them. The three dimensions of vulnerability were addressed in this stage. The four-step interviews were based directly on our conceptual framework, since we asked each farmer to describe the following information: (i) pressures related to climate change (P) and impacts (I) the farmer observes on the farm; (ii) ways to measure the state of the cropping system (S); (iii) adaptation strategies the farmer implements or wishes to implement (R) and

(iv) farm and farmer resources required to implement the adaptations (F). The discussion was facilitated using visual aids (i.e., a board and climate graphs) and the encouragement of the interviewer.

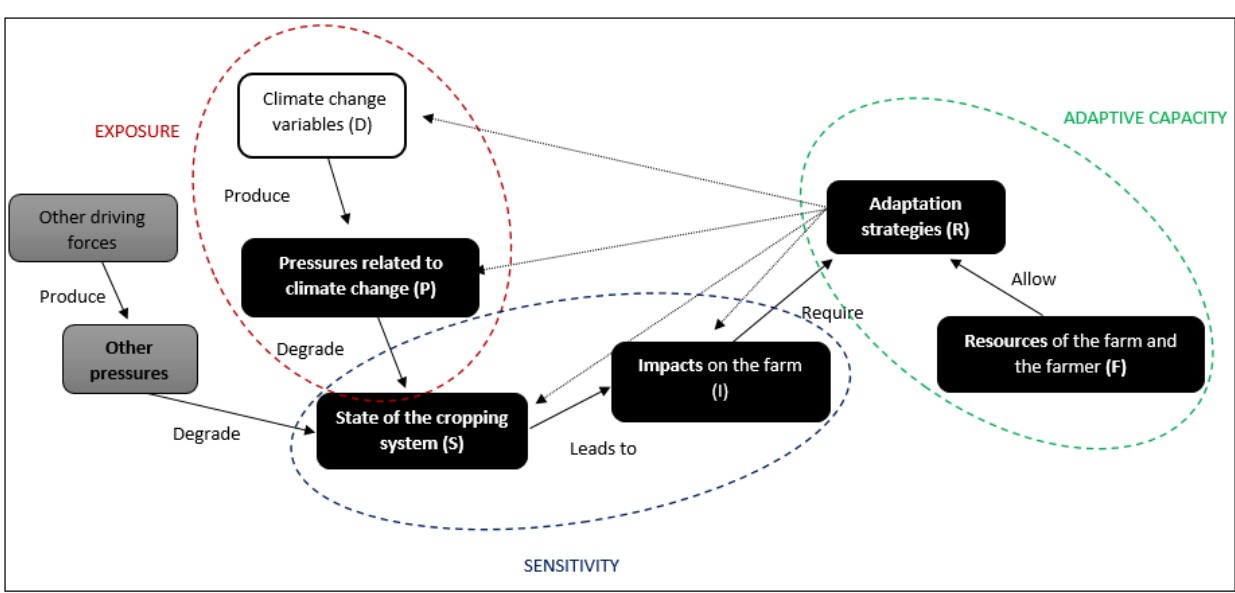

**Figure 1.** Conceptual framework for assessing vulnerability that combines the Drivers (D)–Pressures (P)–State (S)–Impacts (I)–Responses (R) model and the concept of vulnerability.

We designed the questionnaire for the second stage based on the results of the first stage. The first survey revealed a diversity of adaptation strategy criteria elicited from expert farmers. Thus, the second stage of the survey focused on the "adaptation strategies" dimension of our conceptual framework. We tested the genericity of the adaptation strategy criteria obtained in the qualitative survey by asking a larger sample of farmers to select criteria and indicate their relevance. To this end, we created a card game, with each card representing one adaptation strategy criterion (e.g., reduce soil tillage). We then used statistical analysis to explain farmers' opinions via explanatory factors such as the characteristics of the farm and farmer.

**Table 1.** Characteristics of the qualitative and quantitative stages of the survey.

|  | Qualitative Stage | Quantitative Stage |
|---|---|---|
| Objective | Elicit criteria for assessing vulnerability | Test the genericity of the adaptation strategy criteria and explain farmers' opinions |
| Participants | Expert farmers (13) | Farmers representative of the Occitanie region (32) |
| Survey method | Semi-structured interviews on the farm | Semi-structured interviews on the farm |
| Materials | Poster/board representing farm resources; climate graphs | Cards for adaptation strategy criteria |
| Conceptual framework dimensions | All dimensions (i.e., exposure, sensitivity and adaptive capacity) | Adaptive capacity |
| Analysis method | Monography; expert classification | Data cleaning; statistical analysis (regressions, clustering) |

### 2.3. Case Study

The study was conducted in the Occitanie region in southwestern France, within a 100 km radius around the city of Toulouse. Southwestern France is known for its maize production, as it supplies 39% of the national production of the crop [49]. In 2020, Occitanie counted 131,706 hectares of grain maize, including 102,094 irrigated hectares, representing 8% and 15% of the national area of grain maize, respectively [50]. The temperature in Occitanie continues to increase, which has a negative impact on maize cultivation. Higher temperatures in May and June cause problems during reproductive stages, such as pollen degradation and difficulty with absorbing nutrients [5,6]. An increase in summer droughts creates the need for more irrigation water for maize, meaning that maize farms economically depend on irrigation in 6 out of 10 years, on average [49]. Farmers interviewed in the qualitative and quantitative stages came from several departments in the Occitanie region—Tarn, Tarn et Garonne, Gers and Haute-Garonne (Figure 2)—to ensure that we included a diversity of maize farming systems in southwestern France. Access to irrigation differs between the departments, as does soil type, historical types of farm production, climate conditions and the social environment. All farmers interviewed grew irrigated maize (i.e., popcorn, waxy, grain or seed) on at least one field. The Chamber of Agriculture in the region provided contacts for farmers for both stages of the survey.

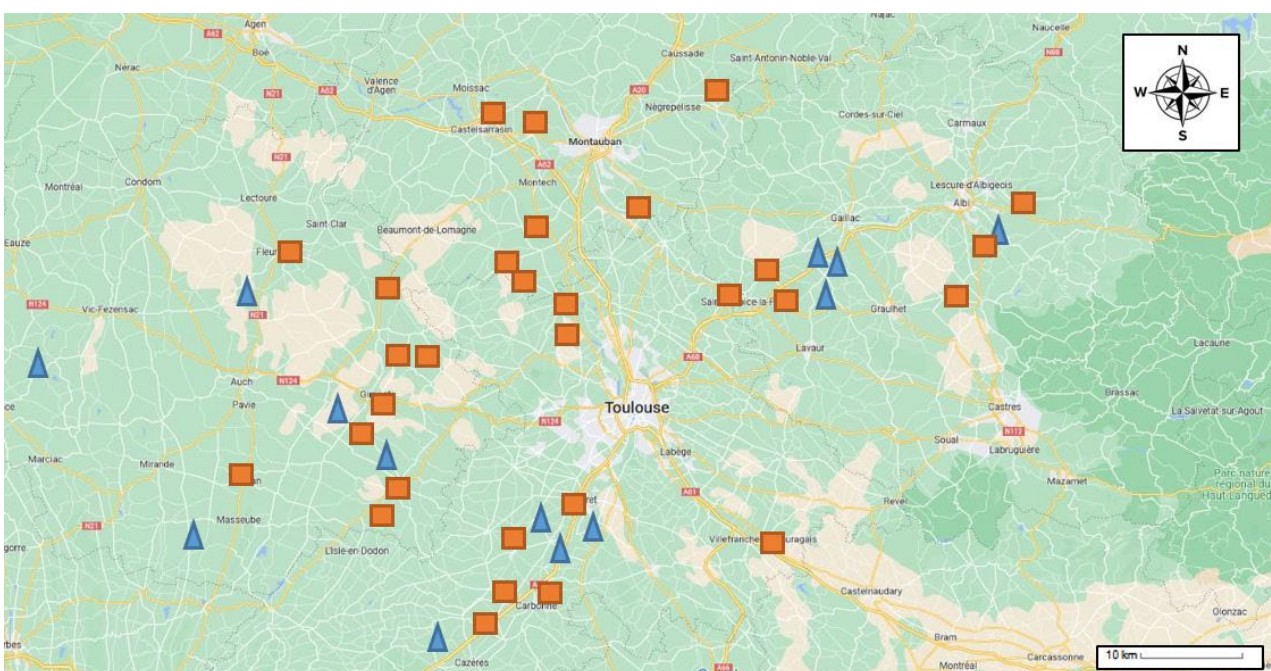

**Figure 2.** Location of the farms of the maize growers interviewed during the first, qualitative stage (initial surveys) (blue) and the second, quantitative stage (deeper surveys) (orange). Map data: 2020, Google.

To elicit criteria in the qualitative stage, we recruited a small sample of expert farmers (13, all male) who were sensitive to climate change and already adopting innovations for adaptation. The interviews, conducted from February to April 2021, lasted 0.75–3 h. For the quantitative stage, we interviewed a sample of 32 maize farmers (all male) representative of the study area in terms of areas and types of production. The interviews, conducted in June and July 2021, lasted 1–3 h.

### 2.4. Data Collection

For the qualitative stage of the survey, two visual aids were used to support the interviews. First, the interviewer showed climate graphs and discussed them with the

farmer, and then encouraged the farmer to express his opinions about the climate pressures and impacts he observed on his farm. After discussing the graphs, the farmer was asked to observe a board representing components of the farming system (Figure 3) and explain how climate change manifests and influences each component. The facilitator encouraged the farmer to identify sensitive elements in his farming system and adaptations he had implemented (or wanted to implement). For each adaptation the farmer identified, the facilitator asked which resources it required. At the end of the interview, the facilitator asked the farmer to summarize the strengths and weaknesses of his farming system in relation to climate change. All interviews were audio recorded to ensure that data were collected accurately.

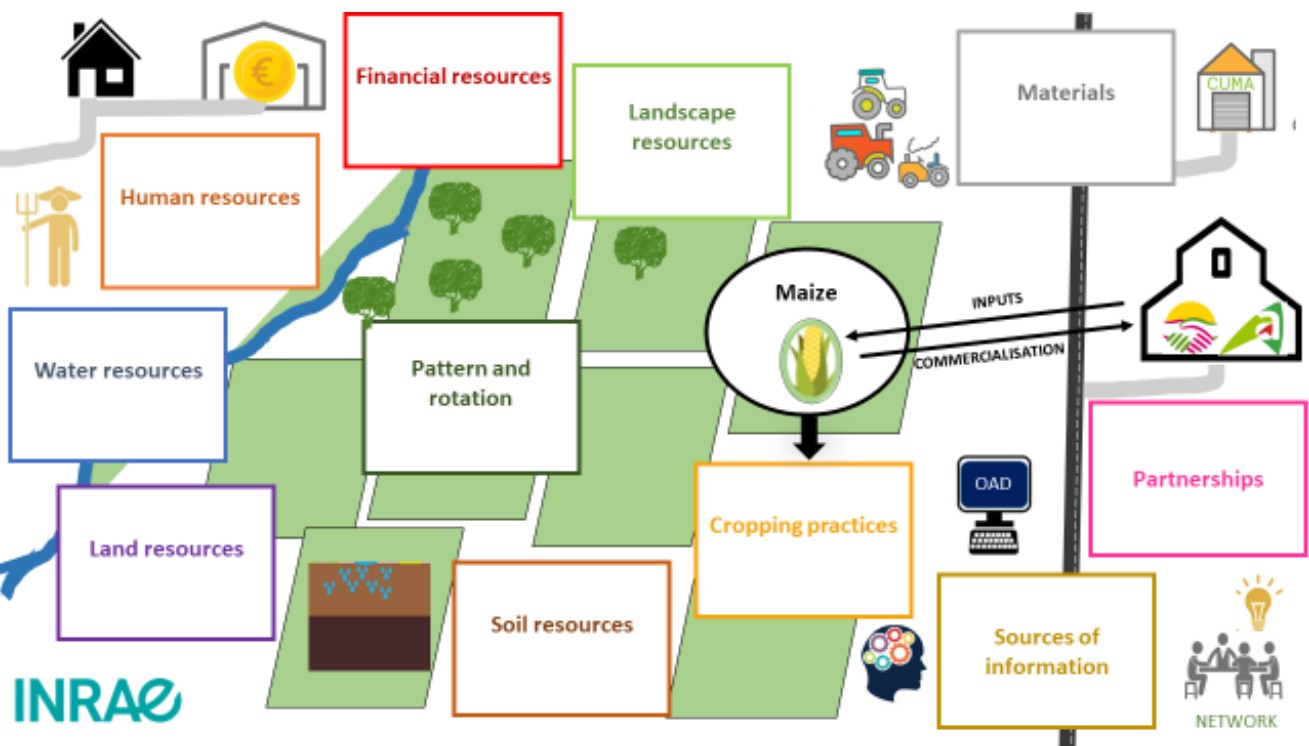

**Figure 3.** The board used in the qualitative survey that represented the components of a farm.

To design the questionnaire for the quantitative survey, we identified a list of explanatory variables and made assumptions about their influence on farmers' opinions about adaptation strategy criteria. We then summarized the extensive list of adaptation strategy criteria obtained in the qualitative stage to (i) identify more general criteria, (ii) avoid redundant criteria and (iii) reduce the number of criteria. We reduced the initial list of adaptation strategy criteria from 50 to 41. The data collection process was divided into three steps (Figure 4a):

- Farmers responded to a questionnaire by telephone to identify the farm characteristics that were potential explanatory variables. The questionnaire was divided into eight categories defined according to expert knowledge, including general information, material resources, water resources, soil resources, financial resources, crops and rotation, human resources and individual resources.
- Farmers were asked to assess the adaptation strategy criteria in a semi-structured interview using the 41 playing cards that represented the adaptation strategy criteria. The interviewer asked the farmer to select the four most relevant cards and the four least relevant cards for reducing vulnerability (Figure 4b).

- The farmer's cognitive and psychological characteristics were identified using a face-to-face questionnaire (Table 2). This questionnaire supplemented the telephone questionnaire by adding new potential explanatory factors.

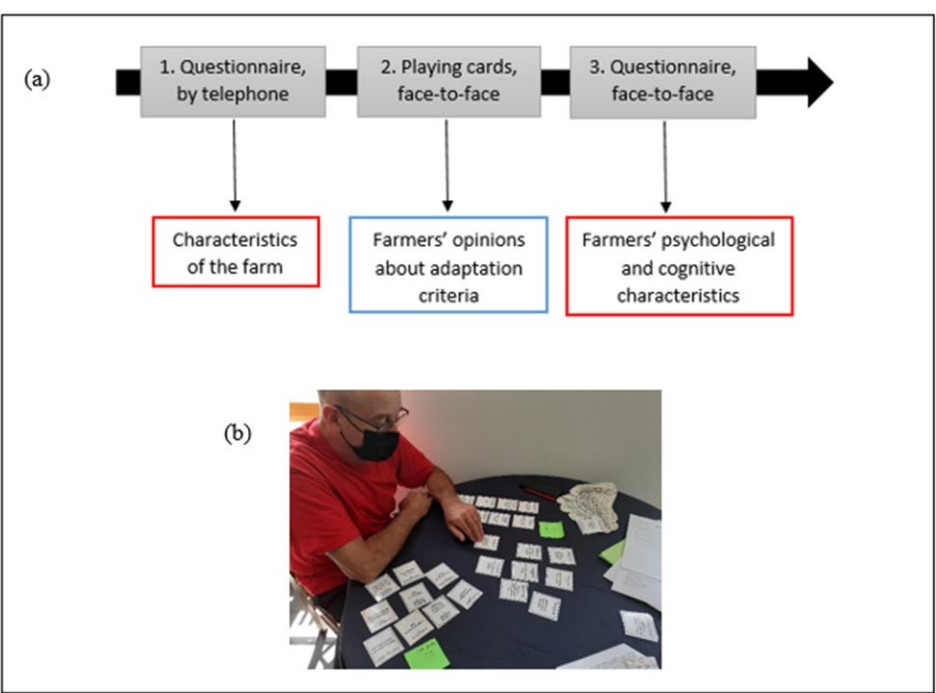

**Figure 4.** (**a**) Main steps of the quantitative survey, showing explanatory variables (red) and the variables explained (blue), and (**b**) a farmer playing the card game used in the quantitative survey.

**Table 2.** Cognitive and psychological variables and the associated elicitation technique in the questionnaire.

| Cognitive and Psychological Factors | Variables | Elicitation Technique in the Questionnaire | Responses Analyzed |
|---|---|---|---|
| Perceptions of climate change | Threat of climate change | Dichotomous question | Yes or no |
| | Level of climate change pressure | Multiple-choice question | Low, medium or high |
| Agroecological practices | Degree of interest in agroecological practices | Self-assessment: score from 1–10 (high interest) | Low (1–3), medium (4–6) or high (7–10) |
| Resistance to change | Degree of attachment to the production system | Self-assessment: score from 1–10 (high interest) | Low (1–3), medium (4–6) or high (7–10) |
| Innovations | Degree of interest in technology | Self-assessment: score from 1–10 (high interest) | Low (1–3), medium (4–6) or high (7–10) |
| Reactivity | Degree of planning | Self-assessment: score from 1–10 (high planning) | Low (1–3), medium (4–6) or high (7–10) |
| Assistance | Favorite information source | Multiple-choice question | Advisors, farmers, technology or laboratory |
| Risk aversion | Degree of risk aversion | Lottery game | High (1–3), medium (4–6) or low (7–10) |

### 2.5. Data Processing and Analysis

For the qualitative survey, each interview was transcribed into monographs. We then identified, via farmers' comments, elements that were related to the dimensions of

the conceptual framework. We collected and classified these elements into a database, and for each element, we determined the associated criteria using agronomic expertise. Finally, we removed redundant criteria and counted the number of times that each criterion was mentioned.

The data collected in the second stage of the survey yielded a database composed of 32 rows (farmers) × 156 columns (explanatory variables) × 41 criteria (explained variables). The statistical challenge was to analyze a database that had many more variables than individuals. Before analyzing the data, we cleaned them in several steps (Appendix A). The cleaning procedure left 25 explanatory variables (Appendix B) for the 32 farmers and 41 explained variables.

To study the distribution of the adaptation strategy criteria among farmers, we developed a typology of these criteria based on farmers' opinions and the number of times they had been mentioned. Classification consisted of multiple correspondence analysis (MCA) followed by hierarchical clustering on principal components (HCPC). We used three additional statistical methods to understand differences in the criteria chosen (Figure 5). We developed regression trees for criteria that were mentioned by at least five farmers. The regression tree method allows for the consideration of local interactions among variables, and is relevant for samples with many variables compared to the number of individuals [6]. We then performed a logistic regression of each criterion and its associated first explanatory variable identified by the regression tree. Finally, we practiced a classification of farmers based on their assessment of the adaptation strategy criteria and studied the distribution of explanatory variables in each cluster. The objective was to create a typology of farmers according to their choice of adaptations, and thus identify explanatory variables for these choices.

The statistical analysis was performed using R software [51], with the MCA and HCPC functions of the FactoMineR package [52] for classifications. The rpart [53] and rpart. plot [54] packages were used for regression trees, and the mlogit package [55] was used for logistic regressions.

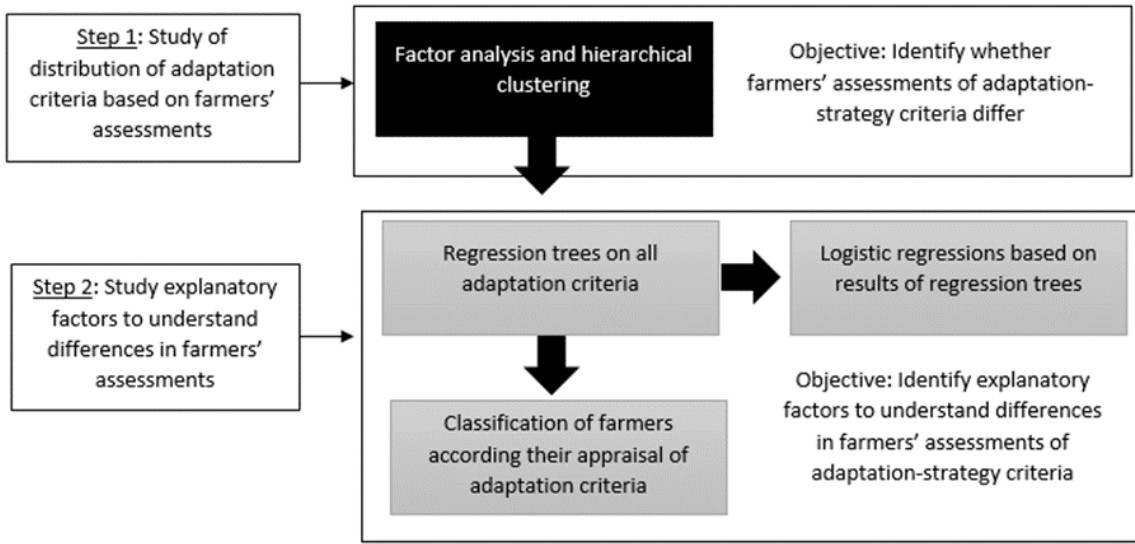

**Figure 5.** Statistical analysis of the quantitative survey.

## 3. Results

### 3.1. Participants

The utilized agricultural area (UAA) varied among the 13 expert farmers involved in the qualitative survey. The mean UAA of 198 ha (standard deviation = 89 ha) was much larger than that of field-crop farms in the region (i.e., 83 ha) [56]. The mean area under

irrigated maize was 52 ha (±32 ha), which was also much larger than that of field-crop farms in the region (i.e., 28 ha).

The mean age and number of labor units of the 32 farmers interviewed in the quantitative survey were representative of field-crop farms in the region. Their farms had a much larger mean UAA and irrigated area (199 and 60 ha, respectively) than those of field-crop farms in the region (99 and 21 ha, respectively). More than 50% of their revenue came from maize production, while the mean was 36% for field-crop farms in the region. A larger percentage of them practiced organic agriculture than the mean percentage of field-crop farms in the region.

### 3.2. Distribution of the Criteria Elicited from Farmer Interviews

The qualitative stage of the survey resulted in a list of 144 criteria distributed among all dimensions of the conceptual framework (Figure 6). The least represented sub-dimension of vulnerability was "farm resources" (nine criteria). Most of its criteria concerned traditional financial assessments of a farm, such as income, gross profit and debt, which all farmers mentioned. The most represented sub-dimension was "adaptation strategies" (50 criteria), which was the most diversified sub-dimension among the 13 expert farmers interviewed.

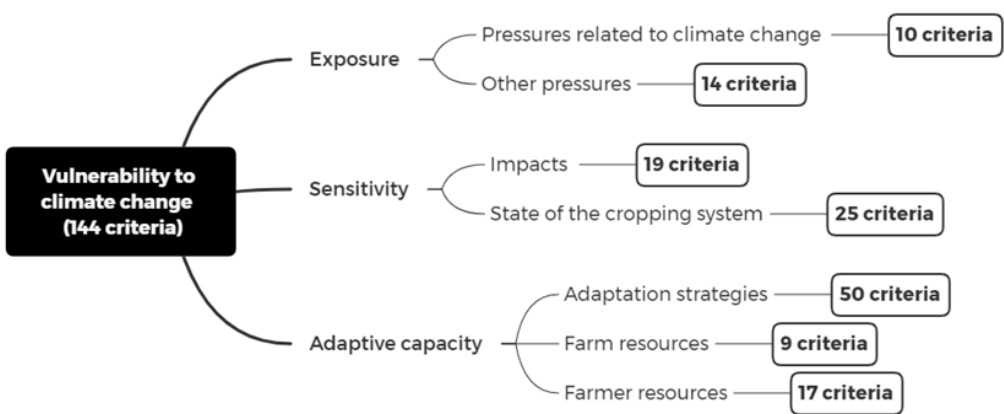

**Figure 6.** Distribution of the criteria in the conceptual framework.

Among the 50 adaptation strategy criteria, only three were mentioned by all farmers: "increase water storage", "diversify water sources" and "irrigate winter crops". All of these criteria were related to water use, which was not surprising in a context of water scarcity. Conversely, 22 adaptation strategy criteria were mentioned only once, such as "partner with a beekeeper", "plant mixed crops" and "introduce livestock". Appendix C shows the number of times that each of the 50 adaptation strategy criteria was mentioned.

### 3.3. Differing Opinions about Adaptation Strategy Criteria among Farmers

The MCA and HCPC of the 41 adaptation strategy criteria according to their relevance for farmers (dimension 1) and the number of times they were mentioned (dimension 2) identified four clusters (Figure 7). Cluster 1 grouped criteria that tended to be considered irrelevant and that were mentioned often, while cluster 4 grouped criteria that tended to be considered relevant and which were mentioned often. Farmers considered the following adaptation strategy criteria to be particularly irrelevant for reducing vulnerability to climate change: "introduce livestock production unit" (A14), "convert to organic farming" (A19), "buy new land" (A17) and "stop growing maize" (A7). Criteria considered particularly relevant for reducing vulnerability to climate change included "use sensors" (A38), "use a maize variety resistant to hydric stress" (A11), "return harvest residues to the soil" (A27), "increase irrigation efficiency" (A34) and "plant cover crops" (A3). A11, A27 and A34 concerned technology or strategies for managing water, while A27 and A3 concerned strategies and crop practices for managing the soil. Cluster 2 grouped criteria that were

mostly irrelevant and rarely mentioned (A10, A18, A23, and A29) along with criteria that generated highly contrasting opinions and were mentioned often (A22, A31). For example, farmers disagreed greatly about the criteria "stop plowing" (A22) and "build or develop a reservoir" (A31). Some farmers considered "stop plowing" a solution to improve soil fertility and control erosion (mentioned as relevant seven times), while other farmers considered it a high risk for crop productivity due to potential problems with weed management and crop establishment (mentioned as irrelevant six times). Cluster 3 grouped criteria that were rarely mentioned and about which farmers' opinions differed, such as "use mechanical weeding" (A24), "practice strip tillage" (A26), "practice direct seeding" (A25) and "diversify water sources" (A33). For example, several farmers agreed that direct seeding was a way to maintain soil quality (mentioned as relevant three times), while other farmers mentioned that equipment costs and uncertainty in productivity made this practice too risky for the financial situation of their farm (mentioned as irrelevant three times).

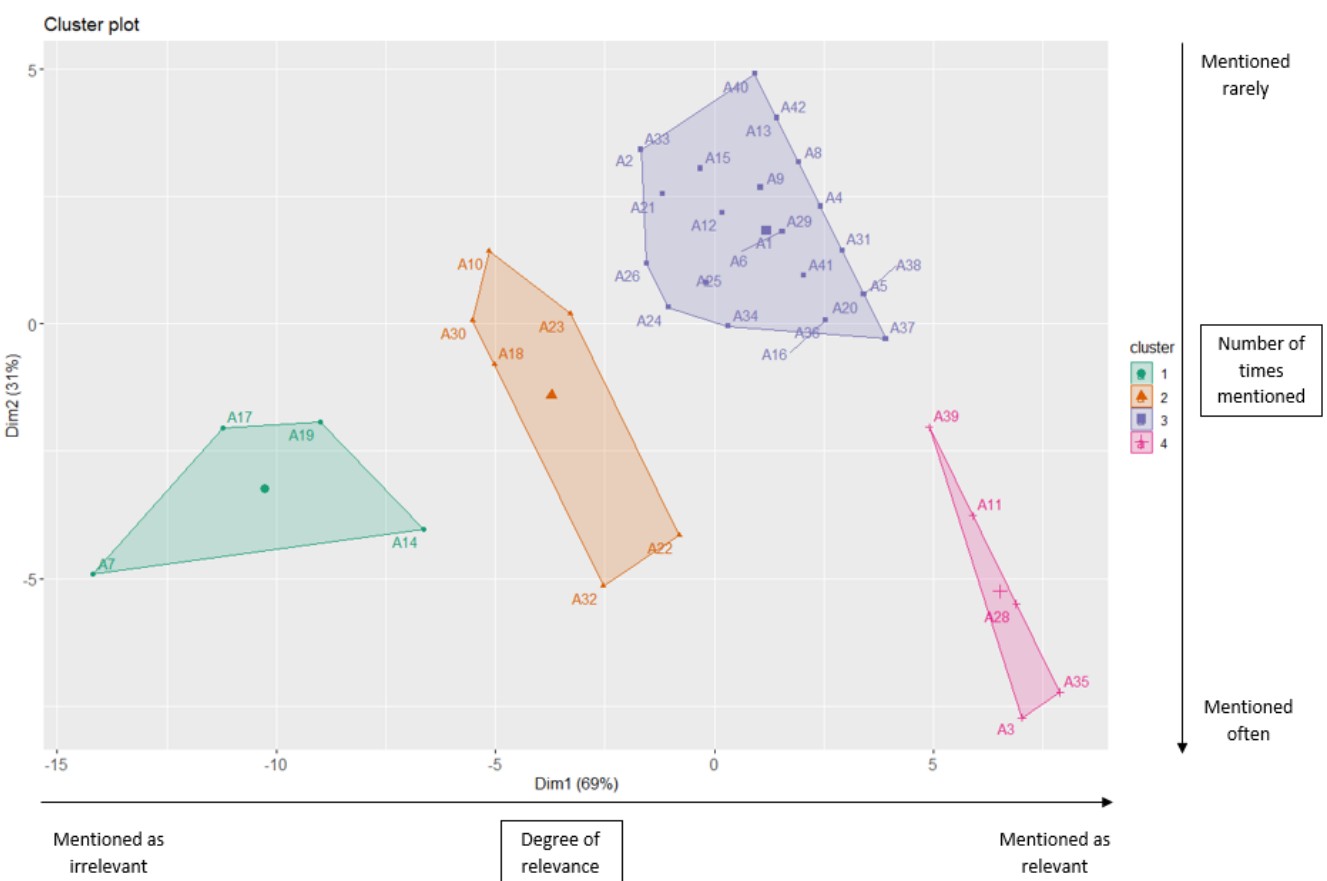

**Figure 7.** Classification of adaptation strategy criteria according to farmers' opinions. Numbers refer to the 41 adaptation strategy criteria (Appendix D).

### 3.4. Explaining Differences in Opinions about the Adaptation Strategy Criteria

For six criteria, logistic regressions confirmed the significant ($p < 0.05$) influence of the first explanatory variable revealed in the regression tree (Table 3). Of the six explanatory factors identified, three were related to farmer characteristics (i.e., age, perceptions of climate change risks, and interest in agroecology) and the other three were related to farm characteristics (i.e., soil types, number of crops).

**Table 3.** Results of regression trees and logistic regressions, with detailed results from logistic regressions. *** $p < 0.001$; ** $p < 0.05$; * $p < 0.1$.

| Criterion | Assessed Relevance | First Explanatory Variable in the Regression Tree | Logistic Regression Results | |
|---|---|---|---|---|
| | | | Direction | Significance |
| **A11: variety resistant to hydric stress** | Relevant | Perceptions of climate change risks: low | + | 0.0368 ** |
| **A27: return harvest residues to the soil** | Relevant | Silty clay soil | + | 0.0292 ** |
| **A29: buy more irrigation quota** | Irrelevant | 1–3 crops<br>4–6 crops<br>(in the rotation) | -<br>- | 0.03279 **<br>0.00324 *** |
| **A31: build or develop a reservoir** | Irrelevant | "Boulbènes" soil (clayey sand) | + | 0.0184 ** |
| **A34: increase irrigation efficiency** | Relevant | Age > 50<br>Age 35–50 | -<br>- | 0.0207 **<br>0.0601 * |
| **A35: irrigate winter crops** | Relevant | Interest in agroecology: high | - | 0.0334 ** |

To supplement the regression methods, we performed the HCPC of farmers based on their opinions of the adaptation strategy criteria (i.e., relevant, not selected, irrelevant). The four clusters it identified contained significant differences in farmers' opinions about the adaptation strategy criteria (Table 4). For each cluster, we identified the adaptation strategy criteria that characterized the cluster the most (Appendix E) and studied the distribution of the 25 explanatory variables (Table 4). Farmers in cluster 1, who adopted slight adaptation strategies, had a similar medium-sized UAA for the sample. These farmers perceived climate change as a threat but had little interest in agroecological practices, which was consistent with the results of the regressions (farmers with high interest in agroecology did not perceive "irrigate winter crops" as relevant). All farmers in cluster 2, who were associated with intensification strategies, practiced conventional agriculture. All of these farmers used more than 2000 m$^3$ ha$^{-1}$ to irrigate and were highly attached to their production system, which explained why they perceived conversion to organic agriculture as irrelevant, as well as their choice of intensification adaptations. Cluster 3 could not be explained by the structural and cognitive variables tested. Cluster 4 (attend training courses and diversify crops) grouped farms with few labor units and clay-limestone soil. They did not plow, had a high interest in agroecological practices and technologies, and had a moderate attachment to their production system. These farmers grew at least four crops in 2020, which was consistent with their opinion that diversifying crops reduced vulnerability to climate change.

**Table 4.** Distribution of the 20 significant explanatory variables within each cluster. Shaded cells identify variables for which farmers' opinions differed within a cluster. The variables of age, water source, the precocity of the maize variety, reactivity and sources of information were tested but are not shown because they were non-significant (i.e., different in all clusters).

| | Cluster 1: Slight Adjustments | Cluster 2: Intensification Strategies | Cluster 3: Diversification of Activities | Cluster 4: Agroecological Innovations |
|---|---|---|---|---|
| Number of farmers | 2 | 11 | 14 | 4 |
| Relevant adaptations to reduce vulnerability to climate change | A5: Balance summer and winter crops; A36: Irrigate winter crops | A30: Renew irrigation equipment; A31: Build or develop a reservoir | A16: Diversify paid activities | A4: Diversify crops; A40: Attend training courses |
| Irrelevant adaptations to reduce vulnerability to climate change | | A19: Convert to organic agriculture; A24: Perform mechanical weeding | A17: Buy new land; A32: Build or develop a reservoir | A23: Lime the soil |

**Table 4.** *Cont.*

|  | Cluster 1: Slight Adjustments | Cluster 2: Intensification Strategies | Cluster 3: Diversification of Activities | Cluster 4: Agroecological Innovations |
|---|---|---|---|---|
| Utilized agricultural area (ha) | 100–200 (medium size) | | | |
| Production system | | Conventional | | |
| Labor units (full-time equivalent) | 1 | | | <2 |
| Education | Bachelor's degree | | | |
| Volume of water (m³ ha⁻¹) | | >2000 | | <3000 |
| Irrigable area | Non-irrigable fields | | | Non-irrigable fields |
| Soil structure | | | | Clay-limestone |
| Number of pivots | None | | | |
| Volume of grain storage (t) | <1000 | | | |
| Number of paid activities | 2 | | | |
| Percentage of revenue from maize | 30–50% | | | |
| Distance between plots (km) | >14 | | | |
| Number of crops | 4–6 | | | ≥4 |
| Soil tillage | Reduced tillage | | | No plowing |
| Climate change threat | Yes | | | Yes |
| Perception of risks from climate change | Medium to high | | | Medium to high |
| Interest in agroecology | Low | | | High |
| Attachment to production system | | Medium to high | | Medium |
| Interest in technology | Low | | | High |
| Risk aversion | Very high | | | |

## 4. Discussion

### 4.1. Psychological Factors That Influence Opinions about the Adaptation Strategy Criteria

The statistical analysis showed that psychological factors explained some of the differences in farmers' opinions about the relevance of the adaptation strategy criteria. Regressions revealed that farmers who perceived the risks of climate change as low perceived the criterion "variety resistant to hydric stress" (A11) as relevant to reducing vulnerability to climate change. Since these farmers were not threatened by climate change, they perceived that slight adjustments, such as changing the variety, were sufficient. Conversely, farmers who perceived high risks prioritized larger adaptations, such as changing the crop pattern. He et al. [57] demonstrated that risk cognition (i.e., individual perception of risks) has a positive influence on adaptive behavior toward climate change. Willaume et al. [58] also related farmers' perceptions of climate change to the type of adaptations they implemented; regarding the "efficiency–substitution–redesign" transition model, farmers with low risk perception tended to choose "substitution" strategies.

Gbetibouo et al. [39] highlighted that farmers' perceptions of climate change are partly influenced by their access to information. Several studies highlighted the importance of cognition, such as the implementation of information [36,38]. Farmers' involvement in a social network would improve adaptive capacity [59]. However, our results indicated that the variable "favorite information source" was not a factor that influenced farmers' opinions about adaptation strategies.

The regressions also highlighted that farmers with high interest in agroecological practices did not perceive "irrigate winter crops" as a relevant adaptation. These farmers may prioritize the conservation of resources, which is a fundamental principle of agroecology. Overall, "interest in agroecology" and commitment to change are determinants of adaptive capacity [35]. Farmers with low interest in agroecology and technology, and with very high risk aversion, tended to perceive slight adjustments as relevant for reducing vulnerability to climate change. Similarly, farmers who practiced conventional agriculture and were highly attached to their production system prioritized intensification adaptations. Conversely, farmers with a high interest in agroecology and technology, and a moderate attachment to their production system, were more likely to perceive innovative agroecological adaptations as relevant to reduce vulnerability to climate change. This confirms that perceptions of innovation (either agroecological practices or technology) [60], attachment to place [35] and resistance to change [60] are key factors that influence the adoption of strategies. Based on our results, high risk aversion seemed to be associated with slight adjustments. Indeed, high risk aversion is a barrier to adaptive behavior [61,62]. Other studies have demonstrated the influence of risk aversion on adaptive behavior [60,63].

Our results confirm that farmers' perceptions influence their adaptive capacity and thus the vulnerability of their farming systems [35]. Adaptation strategies that farmers implement to address climate therefore vary and depend on their cognitive and psychological profiles. Our study could be replicated with a bigger sample in order to confirm our results. Other factors, such as moral concerns (including environmental awareness), intuition and personality [60,64], can also influence the adaptive capacity of farmers and should be considered in future studies to more fully understand farmers' opinions about the relevance of adaptation strategies.

### 4.2. An Original Method Based on Combined Approaches

Our approach for assessing farm vulnerability is original, since we based the identification of vulnerability properties on farmers' expertise, while predefined property approaches are usually based on the literature and/or scientific expertise [28,29]. Thus, our approach was more likely to yield a comprehensive set of indicators that can be managed and adopted easily by farmers and which are appropriate for the farming context. As Wienroth [65] explained in the "let's RULE" model, an innovation should be reliable, useful and legitimate. Another original aspect of our study is the participative and interdisciplinary approach involving behavioral economics.

Similar to Dessart et al. (2019), who proposed a key theoretical framework for our study, our conceptual framework and the choice of the psychological and cognitive factors investigated do not rely on one specific theoretical framework, but are guided by various theories or models of behavior (such as the theory of planned behavior or the theory of expected utility). Indeed, since there is no unified theory of behavior to date and most theories cover only a certain aspect of decision-making [66], our approach allowed us to gather the different behavioral factors that are fundamental to explaining decision-making and, more specifically, the adoption of coping strategies.

One strength of our survey was its combination of qualitative and quantitative approaches, for which we developed original methods to render the concept of vulnerability operational. In the qualitative stage of the survey, criteria were elicited using climate scenarios and a board that represented major components of a farm (e.g., water resources, the farmer's network, equipment). In the quantitative stage of the survey, using cards to represent criteria made the interview playful and interesting for the farmers interviewed.

For the statistical analysis, using a variety of statistical models (i.e., regression trees, logistic regression and classification) enabled us to obtain robust and complementary results. The regression trees were able to consider non-additive effects, combined effects and interactions [6,67] by sequentially dividing responses according to the most relevant explanatory variabale (i.e., by minimizing the locally explained variance). In comparison, the logistic regressions isolated the effects of each variable [68]. The classification pro-

vided results that were complementary to those of the regressions and thus improved the understanding of farmers' opinions about adaptation strategy criteria.

The novelty of this study regarding our results is that we highlighted the important role of the psychological and cognitive profiles of maize growers in their choice of adaptation strategies to address climate change.

The main disadvantage of our study was the small sample size of the quantitative survey (32 farmers). Although the sample was representative of the region, our results cannot be considered general. However, our goal was to test the genericity of adaptation strategy criteria within a group of farmers and not to describe farmers' opinions about the adaptation strategy criteria in the region. We successfully met this goal, since we demonstrated farmers' differing opinions about the adaptation strategy criteria and identified explanatory variables.

### 4.3. The Need to Reconsider Advising and Support Strategies for Farmers

Our results showed that farmers use different adaptation strategy criteria to assess farm vulnerability. Similar to Jones et al. [44], who demonstrated that resilience is not the same for everyone, we determined that vulnerability is also not the same for everyone. Our quantitative focus on adaptations demonstrates that relevant options depend strongly on individual choice, which indicates the need to challenge the genericity of commonly accepted adaptation strategies in the literature. They are presented as universal, but few studies have focused on the conditions of the success or failure of these adaptations in farming systems [10,69]. Misusing adaptations can negatively impact farming systems [16]. Our results indicate that farmers perceive that certain adaptations might increase the vulnerability of their farms. In this case, several studies mention "maladaptations" [10,69–73], which occur when adaptation decisions are made using inaccurate assumptions or failing to consider the potential negative external effects of adaptations (e.g., the degradation of biodiversity, increase in emissions, new or higher costs that farmers did not consider) [74]. This indicates the need to reconsider advising and support strategies for farmers, since public policies and advisor support are often allocated and applied the same way within a region. More individual advising could ensure that farmers adopt adaptation strategies better, as long as farmers' perceptions are considered when developing the recommendations. To this end, discussions with farmers about the vulnerability of their farms are essential before recommending any adaptation strategy. Based on our results, the adoption of agroecological practices is influenced by farmers' interest in agroecology and perception of the risks of climate change. Therefore, agricultural advising and public policies could improve access to information through specific awareness and training campaigns to increase the adoption of agroecological practices.

### 4.4. From Theoretical Results to More Operational Aspects

Various actions could be performed in order to enhance agroecological transition among the maize farming systems in southwestern France. Regarding agricultural advising, there is a need for individual support and groups of discussions and trainings in order to help farmers improve their knowledge on adaptation strategies. Small groups of farmers could be initiated by institutes and private companies advising farmers. A material support for discussion could be a serious game on adaptation strategies and their effects on farms' vulnerability to climate change. Simulated tests of adaptation strategies in virtual conditions could help farmers make decision to adopt specific adaptations and be aware of issues they could face. Farmers' networks that share values and stakes, and which are specifically dedicated to climate adaptation and managed by the Agriculture Chamber, could be created, as it was in the context of pesticide use, with the Defi Zero Phyto network. These networks could offer workshops and visits on farms in order to share experiences and show agroecological innovations. On-farm experiments could be carried out via collaborations between research institutes and farmers, in order to obtain both viewpoints regarding the relevance of the tested adaptations. Finally, public policies

could enhance the agroecological transition by allocating subsidies for farmers who wish to implement agroecological innovations (such as subsidies to buy cover crop seeds or a direct seeder).

*4.5. Toward an Assessment Method That Includes the Adaptive Capacity of Farmers*

In the context of a vulnerability assessment, the dimension of adaptive capacity regarding farmers' internal resources (i.e., psychological and cognitive) requires a specific focus, whereas existing assessment methods often ignore this aspect of vulnerability. One issue is to consider the diversity of farmers' perceptions when developing a tool based on indicators, since farmers who have the same farming system characteristics will not implement the same strategies if their perceptions differ. Our study showed that assessments of adaptive capacity should include criteria and indicators related to psychological factors such as the perception of the risks of climate change, attachment to the production system and interest in agroecological practices. Farmers' opinions need to be compared to objective viewpoints when considering the relevance of adaptation strategies to reduce vulnerability to climate change. Farmers' opinions about adaptation strategy criteria might not agree with scientists' viewpoints due to differences in knowledge, experience and perceptions. This subjective approach for assessing adaptation strategies should be supplemented by consulting the literature and scientific experts to create a typology of relevant adaptations depending on the farming system. Combined with a method for assessing vulnerability, this typology would help advisors provide farmers with effective adaptations, which would result in the development of resilient farming systems. The criteria identified in our qualitative survey related to sensitivity and exposure should also be compared to scientific knowledge to design a multicriteria method for assessing vulnerability to climate change.

**5. Conclusions**

Our study rendered the concept of vulnerability operational by comprehensively identifying its determinants. We developed a conceptual framework that combines the DPSIR model and the concept of vulnerability defined by the IPCC. We identified criteria that farmers use to assess the vulnerability of their farming systems. The criteria of adaptation strategies are diverse and differ among farmers. The statistical analysis using complementary methods showed that farmers' opinions about adaptation strategies are influenced by structural factors (e.g., soil structure), the characteristics of the farmer (e.g., age), the cropping system (e.g., number of crops), as well as cognitive and psychological factors such as risk aversion, attachment to the production system and the perceptions of the risks of climate change. Our study highlighted that farmers' opinions of adaptation strategies are not general, even within the same region. This implies that agricultural advising should be more individualized. The results confirmed our hypothesis that farmers' cognitive and psychological resources influence their adaptive capacity. The relevance of adaptations does not only depend on agronomic or economic performance, but also on farmers' perceptions. Therefore, futures studies dealing with the performance assessment of adaptations should include indicators regarding the level of congruence with farmers' psychological and cognitive profiles. Finally, our study helped to understand farmers' perceptions of the vulnerability assessment, which is the initial step in designing a method to assess the vulnerability of maize farming systems to climate change. Future studies should compare farmers' perceptions of adaptation strategies, sensitivity and exposure criteria to scientific expertise in order to develop a set of indicators.

**Author Contributions:** Conceptualization and methodology, M.A., S.C., J.-E.B. and M.W.; writing—original draft preparation, M.A., S.C. and J.-E.B.; investigation, M.A.; formal analysis, M.A.; writing—review and editing, M.A., S.C., J.-E.B. and M.W.; funding acquisition, M.W. All authors have read and agreed to the published version of the manuscript.

**Funding:** This research was funded by INRAE as part of the VACCARM project of the ACCAF metaprogram, project under S2I 00000285, MP-P10182.

**Informed Consent Statement:** Informed consent to participate and publish was obtained from each participant in the study.

**Data Availability Statement:** Survey instruments and codes used in this study are available from the authors upon request.

**Acknowledgments:** The authors thank Hélène Raynal, who helped design the statistical analysis, and the trainees who helped collect data. The authors also thank the English proofreading reviewers.

**Conflicts of Interest:** The authors declare no conflict of interest. The funders had no role in the design of the study; in the collection, analysis or interpretation of data; in the writing of the manuscript; or in the decision to publish the results.

## Appendix A

The data were cleaned in four major steps (Figure A1). We removed (i) 56 explanatory variables with low heterogeneity (e.g., sex), (ii) 46 explanatory variables that were poorly understood or confusing during the interviews and (iii) 10 strongly correlated variables. In the last step, we sorted the 46 remaining explanatory variables into the eight categories that were previously identified (i.e., general information, water resources, soil resources, human resources, financial resources, crops and rotation, material resources, individual resources) and selected no more than four variables per category to keep those that were the most general. This final step left 25 explanatory variables for the 32 farmers and 41 explained variables.

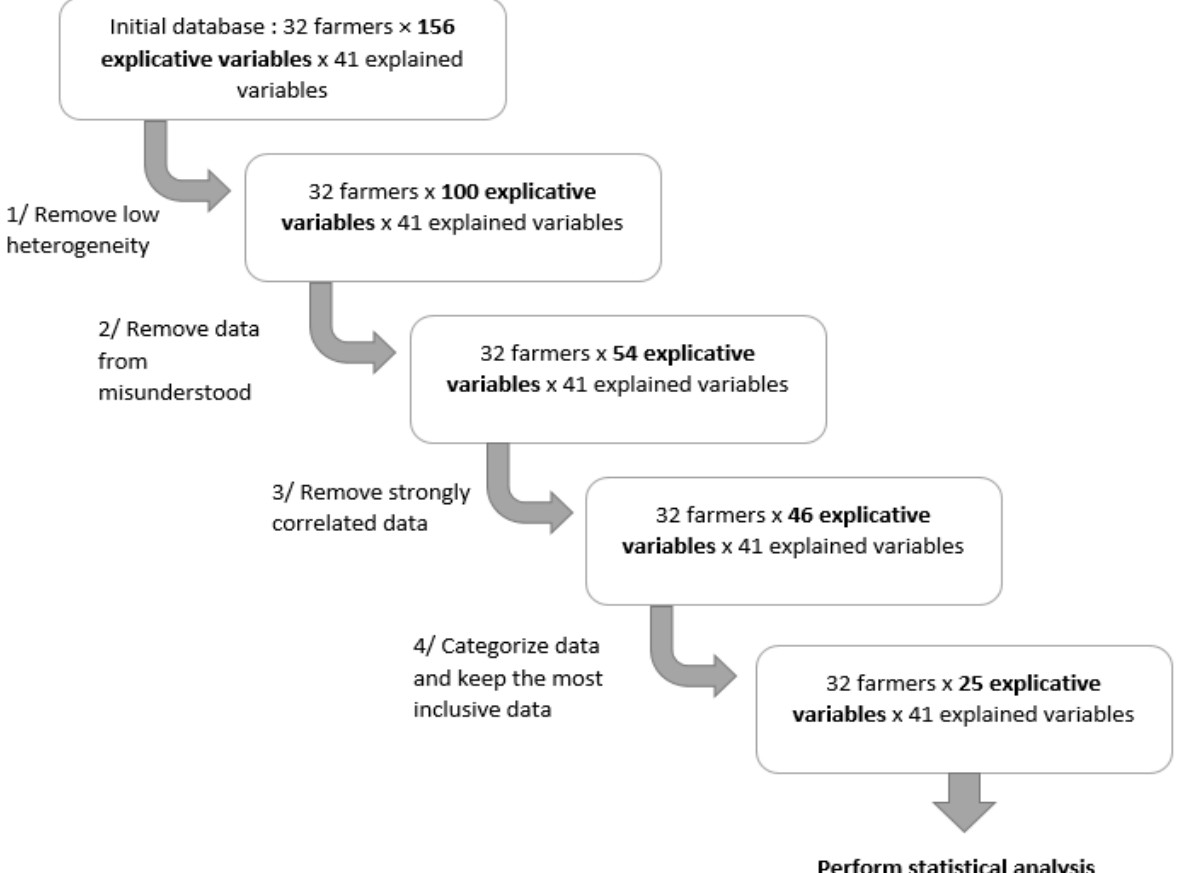

**Figure A1.** Diagram of the data cleaning procedure.

## Appendix B

Description of the 25 explanatory variables.

**Table A1.** Structural and material variables in the questionnaire.

| Category | Variable | Values |
|---|---|---|
| General information | Utilized agricultural area | Small_farm (<100 ha); medium_farm (100–200 ha); large_farm (200–300 ha); very_large_farm (>300 ha) |
| | Production system | All conventional; Organic |
| Human resources | Labor units (LU) | 1_LU; 2_LU; 3_and_more |
| Individual resources | Education | CAP_BEP; Bachelor; Engineer_Masters; Education_Other |
| | Age | <35; 35–50; >50 |
| Water resources | Type of water source | Watercourse; Lake; Well; Watercourse_lake; Watercourse_lake_well |
| | Volume for maize irrigation in 2020 | <2000 m$^3$; 2000–3000 m$^3$; >3000 m$^3$ |
| | Irrigable area | All_irrigable; Fields_non_Irrigable |
| Equipment resources | Number of pivots on the farm | 0_pivot; 1_3 Pivots; >3Pivots |
| | Volume of grain storage | None; <1000 T; 1000–2000 T; >2000 T |
| Soil resources | Soil type | AC (Clay-limestone); AL (Clay-loam); SL (Sandy loam); B (Boulbènes); Others |
| Financial resources | Number of paid activities | 1; 2; 3 |
| | Percentage of revenue from maize in 2020 | <15%; 15–30%; 30–50%; >50% |
| Crops and rotation | Distance between fields | <1 km; 1–5 km; 6–14 km; >14 km |
| | Number of crops in 2020 | 1–3; 4–6; >6 |
| | Soil tillage for maize | Deep tillage with inversion; Deep tillage without inversion; Reduced tillage (including no tillage) |
| | Precocity index for maize | IP_Early (<400); IP_Late (>400); IP_Unknown |

## Appendix C

**Table A2.** Number of times each of the 50 adaptation criteria elicited in the qualitative survey was mentioned. Total number of mentions: 170.

| Criterion | Number of Mentions |
|---|---|
| Increase water storage | 13 |
| Diversify water sources | 13 |
| Irrigate winter crops | 13 |
| Plant cover crops | 12 |
| Stop plowing | 11 |
| Use sensors | 10 |
| Use a weather station | 7 |
| Extend the rotation | 6 |
| Advance the sowing date for maize | 6 |

**Table A2.** *Cont.*

| Criterion | Number of Mentions |
|---|---|
| Diversify paid activities | 6 |
| Self-sufficiency in water | 5 |
| Reduce the precocity for maize | 5 |
| Diversify crops | 4 |
| Partner with a livestock farmer to obtain manure | 4 |
| Buy new land | 3 |
| Buy more irrigation quota | 3 |
| Diversify production | 3 |
| Balance winter and summer crops | 3 |
| Reduce soil tillage | 3 |
| Buy new seeding equipment | 2 |
| Convert to organic agriculture | 2 |
| Practice mechanical weeding | 2 |
| Increase irrigation efficiency | 2 |
| Hire an employee | 2 |
| Use modulation | 2 |
| Shorten water turns | 2 |
| Return harvest residues to the soil | 2 |
| Modify the irrigation strategy | 2 |
| Improve grain storage | 1 |
| Reduce the frequency of field operations | 1 |
| Stop growing maize | 1 |
| Plant mixed crops | 1 |
| Increase grain storage | 1 |
| Advance the date of the first irrigation | 1 |
| Lime the soil | 1 |
| Bury reels for irrigation | 1 |
| Scaring practices for wells | 1 |
| Practice green tillage | 1 |
| Join a group of employers | 1 |
| Plant legume crops | 1 |
| Use decision-support tools | 1 |
| Use a decision-support tool for irrigation | 1 |
| Partner with a beekeeper | 1 |
| Introduce a livestock production unit | 1 |
| Practice sylviculture | 1 |
| Use a specific modulation for fertilizer | 1 |

**Table A2.** *Cont.*

| Criterion | Number of Mentions |
|---|---|
| Use a tall maize variety | 1 |
| Use a maize variety resistant to hydric stress | 1 |
| Practice direct selling | 1 |
| Sell stored maize before summer | 1 |

## Appendix D

**Table A3.** The 41 selected adaptation strategy criteria for the quantitative stage of the survey.

| | | |
|---|---|---|
| **Crop pattern and rotation** | Extend the rotation | A1 |
| | Plant mixed crops | A2 |
| | Plant cover crops | A3 |
| | Diversify crops | A4 |
| | Balance winter and summer crops | A5 |
| | Plant legumes | A6 |
| **Maize cultivation** | Stop maize cultivation | A7 |
| | Advance the sowing date for maize | A8 |
| | Reduce the precocity for maize | A9 |
| | Use a tall maize variety | A10 |
| | Use a maize variety resistant to hydric stress | A11 |
| **Farm scale strategy** | Improve grain storage and commercialization | A12 |
| | Diversify commercialization modes | A13 |
| | Introduce a livestock production unit | A14 |
| | Make a partnership with a neighboring farmer | A15 |
| | Diversify production units and/or paid activities | A16 |
| | Buy new lands | A17 |
| | Hire an employee | A18 |
| | Convert to organic farming | A19 |
| | Buy new equipment | A20 |
| **Cultural practices** | Reduce the frequency of field operations | A21 |
| | Stop plowing | A22 |
| | Lime the soil | A23 |
| | Perform mechanical weeding | A24 |
| | Practice direct seeding | A25 |
| | Practice strip-tillage | A26 |
| | Return harvest residues to the soil | A27 |
| | Use modulation for inputs | A28 |

**Table A3.** *Cont.*

| | | |
|---|---|---|
| | Buy more quota for irrigation | A29 |
| | Improve/renew equipment for irrigation | A30 |
| | Build or enlarge a reservoir | A31 |
| **Water resource** | Advance the date of the first irrigation for maize | A32 |
| | Diversify water sources | A33 |
| | Increase irrigation efficiency | A34 |
| | Irrigate winter crops | A35 |
| | Modify the frequency and/or number of water turns | A36 |
| | Use decision-support tools | A37 |
| | Use sensors | A38 |
| **Sources of information** | Use a weather station | A39 |
| | Attend training courses | A40 |
| | Confront sources of information | A41 |

## Appendix E

**Table A4.** Distribution of criteria within each of the four clusters identified by hierarchical clustering on principal components. See Table 3 in the article for definitions of the criteria codes.

| Cluster | Criteria | Cla/Mod | Mod/Cla | Global | *p* Value | V Test |
|---|---|---|---|---|---|---|
| 1 | A36 relevant | 50 | 100 | 12.90323 | 0.01290323 | 2.486429 |
| | A5 relevant | 50 | 100 | 12.90323 | 0.01290323 | 2.486429 |
| 2 | A32 relevant | 83.33333 | 45.45455 | 19.354839 | 0.0138045121 | 2.462310 |
| | A31 relevant | 100.00000 | 27.27273 | 9.677419 | 0.0367074527 | 2.089003 |
| | A24 irrelevant | 100.00000 | 36.36364 | 12.903226 | 0.0104878436 | 2.559316 |
| | A19 irrelevant | 58.33333 | 63.63636 | 38.709677 | 0.0485031501 | 1.972933 |
| 3 | A32 irrelevant | 100.00000 | 57.14286 | 25.80645 | 0.0003806699 | 3.553134 |
| | A16 relevant | 100.00000 | 28.57143 | 12.90323 | 0.0318131257 | 2.146751 |
| | A17 irrelevant | 76.92308 | 71.42857 | 41.93548 | 0.0037953858 | 2.894685 |
| 4 | A41 relevant | 75.000000 | 75 | 12.903226 | 0.003495948 | 2.920389 |
| | A4 relevant | 60.000000 | 75 | 16.129032 | 0.008580963 | 2.628313 |
| | A23 irrelevant | 100.000000 | 50 | 6.451613 | 0.012903226 | 2.486429 |

Cla/mod: percentage of individuals with the modality inside the class (or cluster). Mod/cla: percentage of individuals of the class (or cluster) with the modality.

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
