# Peer review of "Vulnerability of Maize Farming Systems to Climate Change: Farmers’ Opinions Differ about the Relevance of Adaptation Strategies"

_sustainability, doi:10.3390/su14148275_

Round 1
Reviewer 1 Report
It's innovative method showing the attempt to use knowledge of expert farmers to set as the criteria as well as using both quantitative and qualitative data with statistical analysis. However, in the Materials and Method section, adding detail of data collected for farmer’s cognitive and psychological characteristics by moving Table 1 in Appendix A could help reader understand more clearly. Other minor suggestions are added in the file.

Reviewer 2 Report
The topic is very interesting, it certainly has relevance. The research plan seems well done. The methodology is good, but it needs more robust information in the introduction, analysis, and discussion. However, the relevance to Sustainability should be enhanced in the entire paper with the considerations of scope and readership of the Journal. My major observations are mentioned below
1. The introduction needs to be enhanced with more literature review. There are many climate change impacts on farmers-related studies published recently that need to bring the findings and methods used in those studies in the introduction section.
2. What is the novelty of your work?
3. What is the usefulness of this study? Is this applicable for low and medium-sized cities/towns/villages? Kindly mention the importance of this study in the introduction section
4. The interaction between results & discussion and conclusion needs to be enhanced, which is missing. The information needs to be revised in the conclusion which is already discussed in the discussion section. Need to add new and important information only in the conclusion section which is not been discussed previously.
5. The discussion section needs to be enhanced with more useful recommendations
Reviewer 3 Report
material and methods must be improved.
please add a better version of figure 3.
Reviewer 4 Report
In my opinion, it is a very interesting and well-prepared study. A very comprehensive methodological approach was used here, with detailed explanations. A weakness in such a complex analysis is the small research sample, but the authors are aware of this limitation, so you should accept it. Taking into account that many psychological factors are included in this analysis, it would be worth pointing out the theoretical basis for considering farmers' adaptation strategies (what psychological theory may help explain the behaviour/perception of farmers). I think that it is also worth discussing the theoretical relationship between the vulnerability of farmers and their willingness to adapt new solutions and the sustainability of agricultural systems (farms).
Round 2
Reviewer 2 Report
The authors have done tremendous job in the revised version. I don’t have any additional comments